# Current Perspectives in Liver Transplantation for Perihilar Cholangiocarcinoma

Francesco Giovinazzo [1],[*], Marco Maria Pascale [1], Francesca Cardella [2], Matteo Picarelli [1], Serena Molica [1], Francesca Zotta [1], Annamaria Martullo [1], George Clarke [3],[4], Francesco Frongillo [1], Antonio Grieco [1],[5] and Salvatore Agnes [1],[5]

[1] General Surgery and Liver Transplant Unit, Fondazione Policlinico Universitario Agostino Gemelli IRCCS, 00168 Rome, Italy
[2] Surgical Oncology of Gastrointestinal Tract Unit, Vanvitelli University, 80138 Naples, Italy
[3] Liver Unit, Queen Elizabeth Hospital Birmingham, Birmingham B15 2TH, UK
[4] Centre for Liver and Gastrointestinal Research, Institute of Immunology and Immunotherapy, University of Birmingham, Birmingham B15 2TH, UK
[5] Department of Medicine and Translational Surgery, Università Cattolica del Sacro Cuore, 00168 Rome, Italy
[*] Correspondence: francesco.giovinazzo@policlinicogemelli.it

**Abstract:** Cholangiocarcinoma (CCA) encompasses all malignant neoplasms arising from the epithelial cells of the biliary tree. About 40% of CCAs are perihilar, involving the bile ducts distal to the second-order biliary branches and proximal to the cystic duct implant. About two-thirds of pCCAs are considered unresectable at the time of diagnosis or exploration. When resective surgery is deemed unfeasible, liver transplantation (LT) could be an effective alternative. The overall survival rates after LT at 1 and 3 years are 91% and 81%, respectively. The overall five-year survival rate after transplantation is 73% (79% for patients with underlying PSC and 63% for de novo pCCA). Multicenter case series reported a 5-year disease-free survival rate of ~65%. However, different protocols, including neoadjuvant therapy, have been proposed. The scarcity of organ availability represents a crucial limiting factor in recommending LT preferentially in treating pCCA. Living donor transplantations and marginal cadaveric allografts have proven to be exciting options to overcome organ shortage. Management of jaundice and cholangitis is still challenging for these patients and could impact LT listing. Whether to adopt surgical resection or LT as standard-of-care in pCCA is still a matter of debate, and more prospective studies are needed.

**Keywords:** cholangiocarcinoma; liver transplantation; biliary tumour; review; bile duct neoplasm

## 1. Introduction

Cholangiocarcinoma (CCA) is a rare type of cancer that encompasses all malignant neoplasms arising from the epithelial cells of the biliary tree. It is more prevalent in older populations, and common symptoms of CCA include abdominal pain, jaundice, weight loss, fatigue, and fever. However, these symptoms can also indicate other conditions, making diagnosing CCA in its early stages challenging [1,2].

These are rare and aggressive tumours, with an increased reported incidence over the past few decades (1.2 per 100,000) [3]. CCAs represent ~1% of the total carcinomas, amounting to nearly 3% of all gastrointestinal malignancies. Their prevalence is more marked in older populations aged 50–70 years, with a male-to-female ratio of 2:1. Risk [4].

As CCAs arise from cholangiocytes, they can spread across any tract of the biliary epithelium. Therefore, it is possible to distinguish three subtypes of CCA, depending on their anatomical localisation (Figure 1). About 10% of CCAs are intrahepatic, affecting bile ducts proximal to the second-order biliary branches. About 40% of CCAs are perihilar, involving bile ducts distal to second-order biliary branches and proximal to cystic duct

implant. The remaining 50% of CCAs are extrahepatic, located distal to the cystic duct implant, up to Vater's Ampulla.

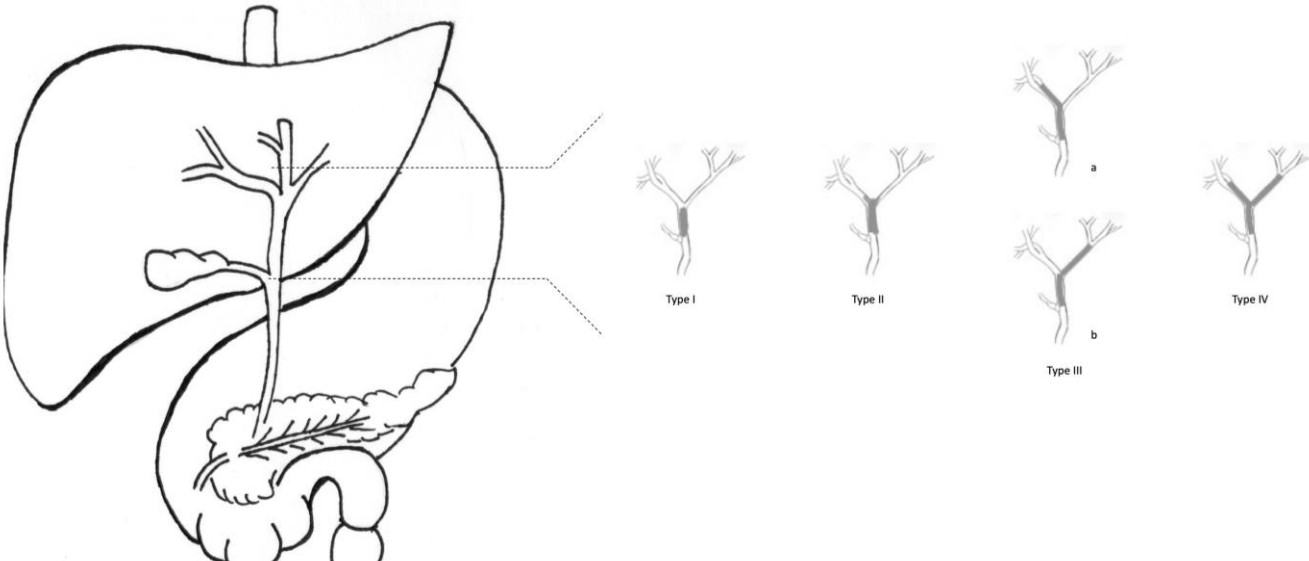

**Figure 1.** Graphic representation of the Bismuth–Corlette classification.

Depending on the disease extension to the main bile duct confluence, it enlists three types of perihilar tumours. CCAs are remarkably aggressive cancers, with a median survival of 10 months when not treated, and an overall life expectancy of fewer than 24 months [5].

The most used tumour marker for CCA is Ca-19.9, with a sensitivity of about 90% and a specificity of up to 98% at a 100 U/mL cutoff [6]. Imaging (contrast-enhanced CT scan, MRI, or MRCP) plays a central role in the diagnostic workup for CCA. Percutaneous transhepatic cholangiography and endoscopic ultrasound are frequently used to define the diagnosis, allowing tissue sampling. A biopsy is not mandatory for fit-for-surgery patients, considering the potential risk of disease seeding and false negatives [7]. However, given that the sensitivity of biopsy is only 69–75%, a significant proportion of patients with pCCA will have a negative biopsy, and a repeat biopsy may delay potentially curative surgery. Patients at high risk of pCCA (no abdominal pain, high CA19.9, and mass on CT) should be considered for surgical exploration even after a negative biopsy. Conversely, patients at low risk of pCCA (abdominal pain, low CA19.9, and no mass on CT) will likely have benign disease. Therefore, that subgroup should undergo at least one repeat biopsy in addition to interval imaging and clinical follow-up [8].

Consequently, pCCA diagnosis requires at least evidence of malignant-like stricture associated with one of the following: positive biopsy/cytology, a mass-forming tissue on cross-sectional imaging, an elevated CA 19.9, or a polysomy by FISH (Fluorescence In Situ Hybridization) [9].

Finally, an 18 F-FDG PET scan could be an adjunct to identify hidden metastases during the staging workup and design the most suitable therapeutic approach [10]. However, 18 F-FDG PET is not recommended by international guidelines [11].

The prognosis of CCA depends on various factors, including the disease's stage, the patient's overall health, and response to treatment. Early-stage CCA has a better prognosis compared to advanced-stage CCA, but the overall survival rate for CCA is still relatively low.

There is no sure way to prevent CCA, but reducing risk factors such as avoiding alcohol abuse, maintaining a healthy diet and lifestyle, and getting regular check-ups can help reduce the likelihood of developing the disease.

## 2. Staging System

Staging of pCCA is essential in determining the best course of treatment.

The American Joint Committee on Cancer TNM stating system, 8th edition, is the most common staging system, which considers the size of the tumour, whether it has spread to nearby lymph nodes, and whether it has metastasised to distant organs, thus providing information on the disease's local and distant extension (Table 1) [12]. Another commonly used staging system is the Blumgart T-staging, which accounts for the anatomical location of pCCA and provides information on hepatic lobar atrophy [13].

**Table 1.** Current American Joint Commission on Cancer staging system for cholangiocarcinoma. TIS = carcinoma in situ; T1 = tumour invades the subepithelial connective tissue; T2 = tumour invades the perifibromuscular connective tissue; T3 = tumour invades adjacent organs. N0 = no regional lymph nodes metastasis; N1 = metastasis to hepatoduodenal ligament lymph nodes; N2 = metastasis peripancreatic, periduodenal, periportal, celiac, and/or superior mesenteric artery lymph nodes. M0 = no distant metastasis; M1 = distant metastasis.

| Stage | T | N | M |
|:---:|:---:|:---:|:---:|
| 0 | is | 0 | 0 |
| I | 1 | 0 | 0 |
| II | 2a-b | 0 | 0 |
| IIIa | 3 | 0 | 0 |
| IIIb | 4 | 0 | 0 |
| IIIc | Any | 1 | 0 |
| IVa | Any | 2 | 0 |
| IVb | Any | Any | 1 |

As pCCAs are rather aggressive and rare carcinomas, they require proper operative treatment to eradicate the disease. There is debate on which approach is best in such cases, but the current standard therapy for pCCA is surgical resection. Therefore, the present review aims to compare available therapeutic alternatives for perihilar CCA (pCCA), focusing on liver transplantation as a promising and increasingly feasible therapeutic option.

## 3. Results of Surgical Resection in pCCA

Complete surgical resection with all gross tumour eradication is the only modality to cure any pCCA. As a rule of thumb, the more distal the malignancy, the more amenable it is to surgical treatment [5]. The surgical strategy for pCCA is en bloc bile duct and gallbladder resection, maintaining a 5–10 mm margin from neoplastic tissue, with locoregional lymphadenectomy. A hepaticojejunostomy with Roux-en-Y reconstruction is required. Furthermore, radical surgical treatment of pCCA implies a hepatic resection, either extended right or left hepatectomy with caudate [14]. When hepatic resection is needed, the surgeon must consider the volume of functioning liver remnant (FLR) to avoid postoperative liver failure. There is no agreed cutoff around the minimum percentage of FLR, but most centres consider 25–30% a safe value in normal liver and 40–50% if cirrhotic. However, it is possible to perform either a portal vein embolisation of the to-be-resected side or the ALPPS procedure (associating liver partition and portal vein ligation for staged hepatectomy) to induce remnant liver hypertrophy. Widespread consensus currently revolves around portal vein embolisation as a standard approach, with ALPPS still feasible in particular cases [15].

About 30% of pCCAs are not eligible for surgery at diagnosis, most of them because they are unfit for surgery or not a resectable disease. An additional 30% prove not surgically treatable following laparotomy/laparoscopic exploration because of metastatic or locally advanced disease [16]. The feasibility of surgical treatment in pCCA depends neither on AJCCs nor Bismuth–Corlette classification. Neither provides information about the local radial spread of cancer, hepatic lobar atrophy, or portal vein invasion. That is why the Blumgart T-staging system has been developed and validated as a tool to assess the resectability of pCCA (Figure 2) [17].

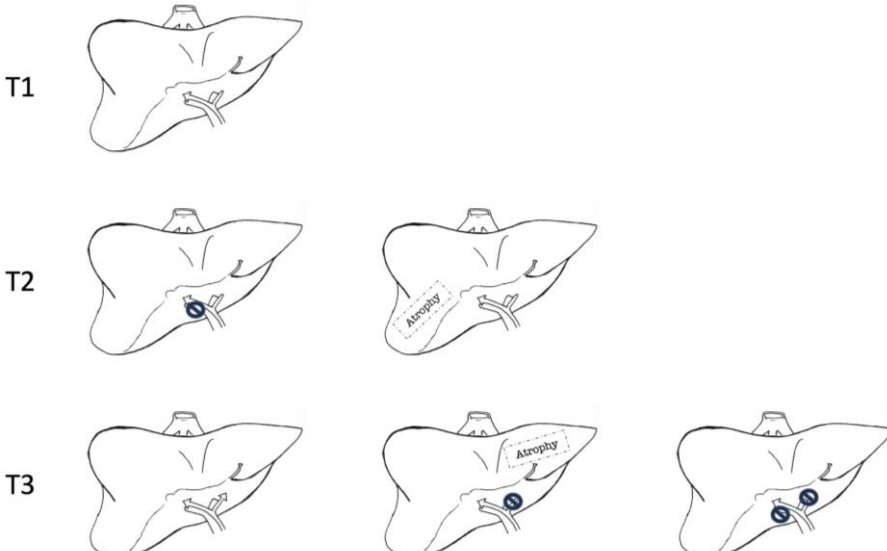

**Figure 2.** Blumgart T-staging system for hilar cholangiocarcinoma. T1 corresponds to a tumour involving biliary confluence with unilateral extension to second-order biliary radicles. T2 corresponds to a tumour involving biliary confluence with unilateral extension to second-order biliary radicles and ipsilateral portal vein involvement or ipsilateral hepatic atrophy. T3 corresponds to tumours involving biliary confluence with bilateral extension to second-order biliary radicles, unilateral extension to second-order biliary radicles with contralateral portal vein involvement, unilateral extension to second-order biliary radicles with contralateral hepatic lobar atrophy, or main or bilateral portal venous involvement.

According to the T-staging system, criteria for non-resectable pCCA are cirrhosis, patient's unfitness for surgery; bilateral disease extension proximal to II grade biliary ducts; portal vein occlusion proximal to its bifurcation; hepatic lobar atrophy with contralateral portal vein occlusion or extension of contralateral disease distal to II grade ducts; unilobar T with contralateral portal vein occlusion; extralocoregional nodal metastases (N2) at histology; and pulmonary, peritoneal or hepatic metastases [6].

The R0-resection rate for pCCA ranges from 50% to 90%, and only a minimal improvement in life expectancy has been registered in such cases. The overall median survival of patients undergoing surgery is 42 months for R0 resections and 21 months for R1 resections [18]. Nevertheless, the 5-year survival rate is significantly different only if R0 resections (up to 60% survival) are compared with non-operated patients, even if it depends on the lymph node status (5-year survival of up to 55% in case of negative lymph nodes vs. 20% in case of positive lymph nodes). There is no difference in 5-year survival rates between patients treated with R1 resections and non-surgically treated patients (less than 10% survival at five years). In other words, the surgical margins, such as lymph node involvement, are an important prognostic factor [19].

Despite primary surgical resection, most pCCA patients are not cured of their disease, as disease recurrence is about 80% within the following eight years, with frequent sites of recurrence in the liver, portal lymph nodes, and peritoneum [20]. For this reason, several centres propose chemoradiotherapy (CRT) not only as neoadjuvant treatment in borderline resectable pCCA (at present, there is no evidence that neoadjuvant CRT is beneficial) [21], but also as an adjuvant approach to enhance surgical results in cases with lymph nodes positive for disease spread [22].

## 4. Adjuvant Therapy

Although, as underlined, resection with curative intent remains the cornerstone of treatment for pCCA [5], it has undergone a continuous evolution in surgical technique and surgical indications [23–27], even in patients undergoing R0 surgical resection and negative

regional lymph nodes, pCCA has a high rate of recurrence and mortality [20,28–30], with 5-year survival ranging from 18 to 54% [5,28–30]. Adjuvant treatment is given after a patient has undergone curative intent surgery to reduce the risk of recurrence and improve survival and can be in the form of chemotherapy or radiotherapy. Its use remains controversial, with conflicting results from different studies. However, some retrospective studies have reported a survival benefit in high-risk patients with positive surgical margins and/or lymph nodes.

The National Comprehensive Cancer Network (NCCN) guidelines recommend adjuvant chemotherapy and/or radiochemotherapy to improve pCCA prognosis [31,32]. Adjuvant therapy for resected biliary tract cancer (BTC) patients with curative intent is capecitabine for 24 weeks (8 cycles), based on the BILCAP study [33]. After resection with curative intent, the role of adjuvant therapy has remained controversial, since the results of two randomised controlled trials (RCTs) have produced other conflicting results. The BILCAP reported an increase in both OS and RFS (recurrence-free survival) in the adjuvant arm; however, the PRODIGE12/ACCORD-18 showed no difference in RFS among patients undergoing surveillance and those who subsequently received gemcitabine and oxaliplatin after resection [33,34]. In the face of this controversy, current NCCN guidelines recommend adjuvant systemic therapy after surgical resection of low-risk CCA versus the surveillance-only option. Some retrospective studies reported a survival benefit in high-risk patients with positive surgical margins and/or lymph nodes [33,34]. As there are contrasting outcomes in low-risk patients, a study by Munir et al. [35] examined the adherence to NCCN guidelines in low-risk resectable CCA, showing improved survival among this group of patients. This study demonstrated that following the evidence-based guidelines was associated with a reduced risk of about 15% of deaths, compared with non-compliance among CCA patients with R0 resection and negative regional lymph nodes [35,36].

The conflicting results from previous studies highlight the need for additional research to establish the most effective and safe adjuvant treatment for this patient population. Further trials are needed to determine the optimal adjuvant therapy for patients with pCCA after curative intent resection. New trials could focus on combination regimens, longer therapy durations, and novel agents, such as immunotherapy, targeted therapy, and personalised medicine. The results of these trials could provide valuable information to guide clinical practice and improve patient outcomes. Additionally, those trials could also examine the cost-effectiveness and quality of life aspects of adjuvant therapy, which would help to inform treatment decisions and policy making.

## 5. Liver Transplantation in pCCA

About one-third of pCCAs are considered unresectable at diagnosis, and another one-third are excluded at the time of exploration because of extensive disease. When resective surgery is deemed unfeasible, liver transplantation (LT) can be an effective alternative [37]. LT allows the treatment of locally advanced disease with complete eradication of the primary tumour and avoids a hypothetical postoperative liver failure due to insufficient remnant liver volume or the impossibility of performing a biliary reconstruction in tumours extended to both right and left ducts or to the secondary order bile ducts [5]. Early experiences in LT in patients with pCCA had a poor outcome, with a 5-year survival rate of ~20%, not significantly different from R1 resection and non-operative management. The main problem identified was an elevated rate of disease recurrence, most commonly found in the allograft or lungs [38].

Therefore, neoadjuvant protocols have been developed to downstage the disease, prevent recurrences and be a bridge strategy to LT. Moreover, neoadjuvant treatment can help identify patients who may not respond to treatment, thereby avoiding futile surgical intervention. Since 2008, the United Network for Organ Sharing (UNOS) has admitted the listing of patients diagnosed with unresectable pCCA as long as a neoadjuvant CRT protocol precedes LT in demonstrated N0-M0 diseases [39].

The first protocol paving the way to LT in pCCA was developed in 1993 by the University of Nebraska and subsequently implemented by the Mayo Clinic [40]. The neoadjuvant therapy described in this study consists of a combination of external-beam radiation, brachytherapy with 5-FU radiosensitising during RT, and subsequent capecitabine administration until LT (Figure 3) [41]. Enrollment criteria adopted were the non-resectability of the tumour, a maximum lesion diameter of 3 cm (several studies showed better prognosis when the tumour size was <3 cm) [42], absence of intrahepatic and extrahepatic metastases, and patients had to be medically fit for both CRT and LT. The exclusion criteria were intrahepatic cholangiocarcinoma, uncontrolled infection, prior radiation or chemotherapy, prior biliary resection or attempted resection, intrahepatic metastases, evidence of extrahepatic disease, history of other malignancy within five years, and transperitoneal biopsy (including percutaneous and EUS-guided FNA).

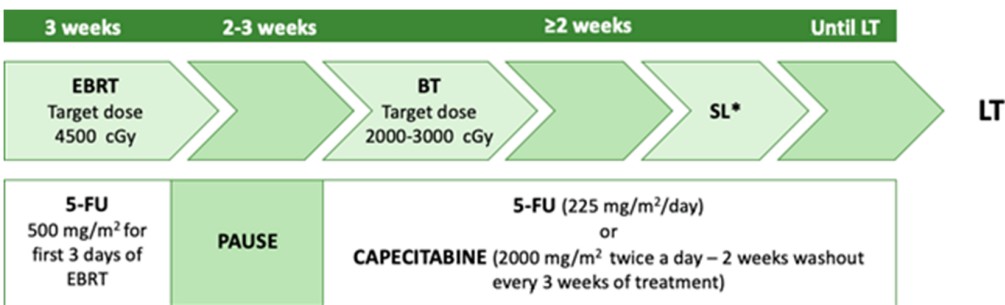

**Figure 3.** Mayo Clinic neoadjuvant protocol. EBRT: external beam radiotherapy; BT: brachytherapy; SL: staging laparotomy; LT: a liver transplant. * with biopsies of at least common hepatic artery and distal bile duct lymph nodes.

The overall survival rates after LT at 1 and 3 years were 91% and 81%, respectively [19]. Five-year survival after transplantation and neoadjuvant was 73%, but 79% for patients with underlying PSC, and 63% for those with de novo CCA [43].

Multicenter case series reported a 5-year disease-free survival rate of ~65%. However, more than two-thirds of the patients had PSC as an underlying disease, compared to about 5% in other pCCA cohorts [44]. Some critical points of Mayo Clinic pre-LT protocol were the staging surgery, which involves, 2–6 weeks after initiation of radiotherapy, a complete abdominal exploration with biopsy of any lymph nodes or nodules suspected of tumour, the examination of the tumour, and routine biopsy of regional lymph nodes (along the distal common bile duct, hepatic artery, and in the celiac and peripancreatic area) [45]. Another criticality was the selection of patients with early-stage pCCA arising in the setting of PSC with negative cytology, negative FISH, and no residual tumour in the specimen after transplantation, putting into question whether the patient ever had cancer. On the other hand, those patients were under close surveillance and had a cancer diagnosis very early, and PSC patients seemed to have the most favourable outcomes [46]. Nevertheless, several centres performing LT for pCCA adopted the Mayo Clinic protocol as the standard of care whenever complete surgical resection was impossible [11,35].

Toronto General Hospital and Princess Margaret Cancer Centre have developed another neoadjuvant regimen [47]. The protocol was modelled on the Mayo Clinic protocol, aiming to implement the radiosensitising phase and the maintenance chemotherapy, while reducing biliary toxicity stemming from brachytherapy (Figure 4).

The exclusion criteria in this study were inability to consent, prior attempted resection within the past 12 months, prior upper abdominal radiation therapy, a trans-peritoneal biopsy of the primary tumour within the past 12 months, prior malignancy diagnosed in the last five years, uncontrolled infection, inability or unwillingness to complete the protocol due to comorbid conditions, or ECOG $\geq$ 3 at an initial consultation. Criteria for exclusion from the protocol were evidence of metastatic disease on follow-up imaging, surgical staging positive for malignancy, or disease progression on neoadjuvant treatment. During

neoadjuvant treatment, patients were reviewed weekly, and followed up one month after surgical staging and three months after transplant. Repeat CT chest, abdomen, and pelvis, and CA19-9 testing was performed three months following the protocol's enrollment until transplant. The median total duration from the first consultation and start of neoadjuvant chemotherapy until LT was 48.6 (40.7–77.9) weeks and 37.0 (30.0–63.7) weeks, respectively. There were no dropouts due to toxicity, although 61% dropped out due to metastatic disease development and medical comorbidity exacerbation. The OS after LT at 1 and 2 years was 83.3% and 55.6%, respectively [31]. However, due to the Toronto protocol's recent validation, a 5-year follow-up is not available at present, nor a comparison with the Mayo Clinic experience.

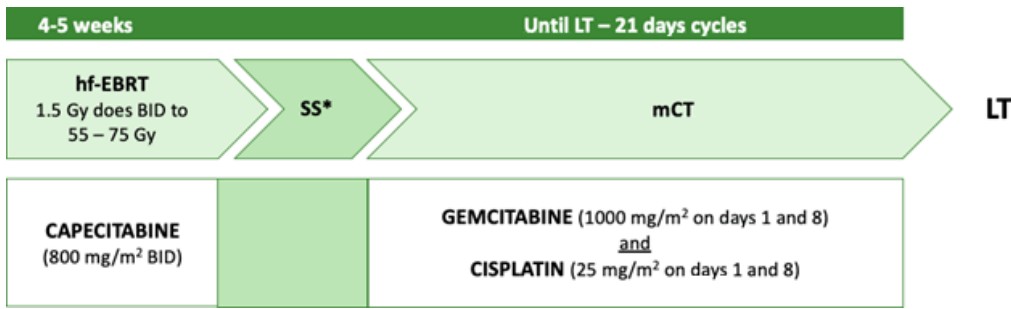

**Figure 4.** Toronto neoadjuvant protocol. Hf-EBRT: hyperfractionated external beam radiotherapy; mCT: maintenance chemotherapy; SS: staging surgery (either laparotomy or laparoscopic); LT: liver transplant; BID: twice a day. * with biopsies of hepatic artery and hepatoduodenal lymph nodes.

In a pilot study in 2014, the University of Michigan designed a neoadjuvant protocol based on stereotactic body radiation therapy (SBRT) [48]. According to the authors, this approach can minimise toxicity on adjacent healthy structures by concentrating radiation doses on the local disease. The inclusion criteria were the same as the Mayo group. After two weeks of SBRT and one week of rest, patients were initiated to the capecitabine schedule and underwent a staging operation. Maintenance chemotherapy was provided until LT (Figure 5). In the reported single-institution experience, histological proven tumour response was detectable in 66% of neoadjuvant therapy patients. A single-centre experience using brachytherapy, external beam radiotherapy and 5-fluorouracil (5-Fu) showed that, in selected patients with unresectable pCCA, long-term survival was 94% and 61% at 1 and 4 years, respectively, with comparable results to other series. However, in-hospital mortality was 20% [49,50].

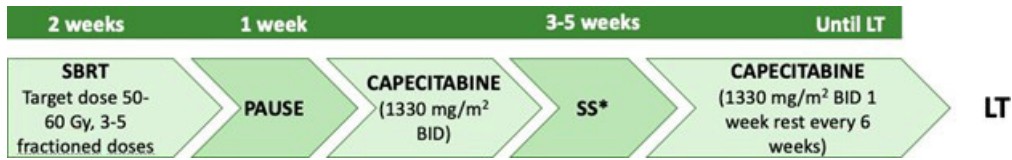

**Figure 5.** University of Michigan neoadjuvant protocol. SBRT: stereotactic body radiotherapy; SS: staging surgery; LT: liver transplant; BID: twice a day. * with lymph nodes biopsies.

There is no evidence that SBRT is effective against pCCA, because of the lack of comprehensive case observations and prospective studies. However, a small sample and mostly retrospective studies combining chemotherapy with SBRT have shown promising results in obtaining an oncologic response [51]. Despite the requirement for a neoadjuvant protocol in transplanting patients with pCCA in many national/international networks, CRT before LT is not mandatory in several liver transplantation programs (e.g., EUROTRANSPLANT criteria, adopted in Austria, Belgium, Croatia, Germany, Hungary, Luxembourg, the Netherlands, and Slovenia) [37]. The rationale, beyond avoiding neoadjuvant CRT, is preventing

systemic toxicity from shortening the waiting time and selecting only LT candidates with no lymph nodal metastases. Even in non-neoadjuvant diseases, staging surgery before LT has a significant role to play, and it has been widely adopted because of the predictive value of nodal metastases. N0 patients not included in any bridge-to-LT CRT protocol have a 5-year survival of more than 50% [52].

## 6. Discussion and Future Perspectives

The decision for surgical resection or LT in pCCA requires a multidisciplinary approach, high-volume centres, and expert hepatobiliary surgeons. For all cases of perihilar cholangiocarcinoma, a multidisciplinary approach is recommended to provide the best possible outcome for the patient. A large body of evidence supports the use of a multidisciplinary approach to managing perihilar cholangiocarcinoma. Similarly, a consensus statement from the International Cholangiocarcinoma Consensus Group emphasised the importance of a multidisciplinary approach, including collaboration between surgeons, medical oncologists, and radiation oncologists, in the management of perihilar cholangiocarcinoma [53].

In several cases, patients diagnosed with pCCA are eligible for surgical resection, and sometimes patients can be selected for LT. However, due to the rarity of the disease and lack of phase 3 RCTs, there is often no clear standard pathway to refer to. The preoperative workup is complex, and the decision-making should balance risks with prognostic factors and expected results. That is why the possibility of comparing different operative strategies and their outcomes becomes pivotal (Table 2) [54].

**Table 2.** Liver transplantation or resection for pCCA.

| Authors | Year | Number of Patients | Procedure | Neoadjuvant | | | Follow up Years, (Range) | Median Survival, Months (Range) | LR (%) | | LT n(%) | |
|---|---|---|---|---|---|---|---|---|---|---|---|---|
| | | | | Protocol | Cholangitis n (%) | Drop-Out n (%) | | | OS n (%) | Disease-Free Survival n (%) | OS n (%) | Disease-Free Survival n (%) |
| Sudan et al. [41] | 2002 | 17 | LT | Nebraska | 5 (29) | 6 (35) | 7.5 (2.8–14.5) | 25 (4–174) | - | - | 5 (45) | 5 (45) |
| Heimbach et al. [45] | 2004 | 56 | LT | Mayo protocol | - | 14 (39) | 3.5 (0.5–10.9) | - | - | - | 11 (50) | 30 (90) |
| Sano et al. [24] | 2006 | 102 | LR | no | - | - | 2.8 (0.4–5.2) | 19 (4–62) | 62 (61) | 60 (50) | - | - |
| Michiaki et al. [25] | 2010 | 125 | LR | no | - | - | 1.5 (0.02–8.25) | 26.8 (nr) | 43 (35) | - | - | - |
| Murad et al. [44] | 2012 | 287 | LT | Mayo protocol | - | 71 (25) | 2.5 (0.1–17.8) | 14.4 (1–205.2) | - | - | 165 (57) | 244 (80) |
| Nagino et al. [23] | 2013 | 574 | LR | no | - | - | 5.8 (nr) | - | 83 (21.4) | - | - | - |
| Duignan et al. [49] | 2014 | 27 | LT | Mayo protocol | - | 7(26) | 3.1 (1.6–6.3) | - | - | - | 9 (45) | 9 (56) |
| Welling et al. [48] | 2014 | 17 | LT | Mayo protocol | 6 (50) | 10 (59) | 1.2 (nr) | - | - | - | * | * |
| Koerkamp et al. [32] | 2015 | 306 | LR | no | - | - | - | 40 (nr) | 145 (30) | 111 (31) | - | - |
| Marchan et al. [50] | 2016 | 10 | LT | Mayo protocol | - | 2 (20) | 2.5 (nr-3.1) | - | - | - | ° | - |
| Ethun et al. [54] | 2018 | 304 | LT/LR | Mayo protocol | - | 29 (41) | - | § | 35 (18) | - | 27 (65%) | - |
| Loveday et al. [47] | 2018 | 43 | LT | Toronto | 12 (67) | 11 (62) | 1.5 (0.5–1.8) | 17.7 (4.9–29.6) | - | - | & | & |
| Vugts et al. [37] | 2021 | 34 | LT | no | - | 32 (94) | - | - | 55 (36) | - | 1 (50) | 1 (50) |
| De Bellis et al. [26] | 2022 | 100 | LR | no | - | - | 3.4 | 40.5 | 36 (36) | 16 (16) | - | - |
| Matsuyama et al. [21] | 2022 | 60 | LR | Gemcitabine S1 | 27 (45) | 17 (28) | 2.5 (0.4–8.9) | 50.1 (nr) | 22 (36) | 33 (55) | - | - |

* survival reported only at 1-year. ° survival reported only at 2-years. § resection (median 17.1 months, 95% CI 17.8–26.3), transplantation (median 77.4 months, 95% CI not reported). & Two-year post-transplant OS 55.6% and DFS 66.6%.

In most case series, peri-operative mortality of surgical resection is around 10% and equates to LT. On the other hand, postoperative complications overall are significantly lower in transplanted patients than resected ones (<50% vs. 68%); nonetheless, there are no

differences between the two groups as far as the incidence of major postoperative complications and postoperative liver failure are concerned (~40% and ~10%, respectively) [19].

No difference in the rate of disease recurrence between surgical and transplant series has been identified; according to a multicenter study, the 5-year overall survival for neoadjuvanted LT happens to be significantly higher if compared to SR (54% vs. 29%) [55,56].

Under these assumptions, LT seems to be a preferable curative choice for pCCA. Though, some additional considerations must be included in the decision-making approach.

To start with, scarcity of organ availability represents a crucial limiting factor in recommending LT preferentially in treating pCCA [57]. In recent years, living donor transplantations and marginal cadaveric allografts have proved an exciting option to overcome organ shortage. The sustainability base principle in transplantation is that the 5-year prognosis must be equivalent to other oncological pathologies currently treated with LT, not to misallocate allograft [58]. Yet, the side effects of life-long immunosuppression after LT are worth some consideration.

Resection should be considered the first-line treatment if the disease is localised. After resection, consider adjuvant therapy (chemotherapy or radiotherapy) could improve outcomes and prevent recurrences. Neoadjuvant therapy may be considered to downstage the disease and make resection possible or liver transplantation. The use of neoadjuvant therapy has also been shown to be effective in some cases of perihilar cholangiocarcinoma. A study published in the BMC Cancer found that neoadjuvant therapy followed by surgery was associated with improved survival outcomes, compared to surgery alone, in patients with locally advanced perihilar cholangiocarcinoma [56]. If the disease is not amenable to surgical resection or if there is evidence of liver failure, liver transplantation could be considered an option.

When analysing the results of various LT protocols for pCCA, they appear to be quite promising. Nevertheless, it is questionable whether this happens because of actual efficacy or somewhat due to some previous selection of patients with a better prognosis. On that note, it is difficult to determine whether LT outcomes in pCCA are partially influenced by the efficacy of neoadjuvant chemoradiotherapy (CRT), as protocols are quite heterogeneous, and there is an absence of direct comparison with surgical resection. Tumour response to CRT schemes is demonstrated in about 60% of pCCA, with histological evidence of tumoural necrosis in explanted livers and the extent of residual tumour predicting outcome in patients with pCCA [59]. Therefore, macroscopic, microscopic, and immunohistochemical analyses show a high rate of residual tumour control in neoadjuvant patients [59]. The 5-year overall survival among those who did not receive LT after CRT is more than 20%, which is a remarkably better result than patients not treated at all, and quite similar to resection outcomes. Again, aggressive CRT protocols may represent a substantial operative challenge and jeopardise other therapeutic chances if unsuccessful or affected by adverse events [60].

In conclusion, adjuvant and neoadjuvant treatments have their advantages and disadvantages. The choice of treatment depends on various factors, such as the stage of the disease, the patient's overall health, and the treatment goals. Whether to adopt surgical resection or LT as standard-of-care in pCCA is still a matter of debate, and more prospective studies are needed.

**Author Contributions:** Study concept and design (F.G.); acquisition of data (M.M.P., F.C., S.M., F.Z.); analysis and interpretation of data (F.G., F.C., M.P., S.M., F.Z., A.M., G.C., M.M.P., F.F., A.G., S.A.); drafting of the manuscript (F.G., F.C., M.P., M.M.P., S.M., F.Z., A.M., G.C., F.F., A.G., S.A.); critical revision of the manuscript for important intellectual content (F.G., F.F., A.G., S.A.); study supervision and project administration (F.G., S.A.). All authors have read and agreed to the published version of the manuscript.

**Funding:** This research received no external funding.

**Conflicts of Interest:** The authors declare that they have no known competing financial interests or personal relationships that could have appeared to influence the work reported in this paper.

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
