# Peer review of "Current Perspectives in Liver Transplantation for Perihilar Cholangiocarcinoma"

_curroncol, doi:10.3390/curroncol30030225_

Round 1

Reviewer 1 Report

Liver transplantation for patient with peri-hilar transplantation seem to be promising, but without enough scientific proof, which can permit to adopt this strategy as a standard of care.

Authors had to moderate affirmation.

Abstract:

Line 20: Authors could replace “can” by “could” in the sentence: “ liver transplantation could be an effective treatment”.
Following international recommendations, liver transplantation could not be considered as a standard treatment.

Introduction:

Could the authors explain polysomy by FISH.

Lines 69-70: Following the ESMO guidelines, FDG-PET is not recommended for the examination of metastatic extension, thoraco-abdomino-pelvic CT remains the reference, and for local extension, MRI is the reference.

Authors had to add the references of international recommendations guiding treatment of cholangiocarcinoma and the element of decision.

Line 92: authors cannot write:” Complete surgical resection with all gross tumour eradication is the best curative option for any CCA”. Complete surgical resection is the only modality that can cure CCA.

Line 176: “The overall survival after LT at 1 and 3 years was 91% and 81%, respectively [18,27]”. Could the authors clarify from which study were these percentages. Because from the Mayo Clinic's study, the overall survival was 88% at 1 year and 82 % 5 years after transplantation.

Line 188-190: “Nevertheless, several centers performing LT for pCCA adopted the Mayo Clinic protocol as the standard of care whenever complete surgical resection was impossible”. Authors had to modulate the affirmation, and add that this strategy is not recommended in international guidelines.

A table could be added to compare the different rates of overall survival between different teams studied liver transplantation, compared to surgery.

Authors had to insist on the heterogeneity on chemotherapy protocol studied on neo adjuvant setting before liver transplatation, and the absence of direct comparison.

Conclusion:

Lines 264-266: Authors cannot conclude with this affirmation, without any data from a randomized phase 3 trial.

Lines 288-292: Authors described loco-regional treatment; they could develop this paragraph. Authors discussed neo adjuvant treatment before surgery, but in the text, authors did not described studies on surgery. They could add a paragraph with main studies of surgery and the neo adjuvant strategy.

Author Response

Dear Professor/Doctor,

Thank you very much for your suggestion and recommendation. Please see below the answer to your comment with the realtive editings. All the change have been highlithed in yellow. That has helped to improve the quality of the manuscript.

R: Liver transplantation for patient with peri-hilar transplantation seem to be promising, but without enough scientific proof, which can permit to adopt this strategy as a standard of care. Authors had to moderate affirmation.

Authors: Thank you very much for the comments. The authors have moderated the statement in the new version.

R: Abstract:

Line 20: Authors could replace “can” by “could” in the sentence: “ liver transplantation could be an effective treatment”.
Following international recommendations, liver transplantation could not be considered as a standard treatment.

Authors: Thank you very much. In the new version, “can” has been replaced by “could” in the sentence: “ liver transplantation could be an effective treatment”.

R: Introduction:

Could the authors explain polysomy by FISH.

Authors: Thank you very much FISH is now written in short and extended form.

R: Lines 69-70: Following the ESMO guidelines, FDG-PET is not recommended for the examination of metastatic extension, thoraco-abdomino-pelvic CT remains the reference, and for local extension, MRI is the reference.

 Authors: Thank you very much. The statement has been changed as follows:

“Finally, (18)F-FDG PET scan could be an afjunct to identify hidden metastases during staging workup and design the most suitable therapeutic approach [10]. However (18)F-FDG PET is not recommended by international guidelines [11].”

R: Authors had to add the references of international recommendations guiding treatment of cholangiocarcinoma and the element of decision.

 Authors: Thank you very much. Reference 31 has been added and refers to the NCCN Clinical Practice Guidelines in Oncology

R: Line 92: authors cannot write:” Complete surgical resection with all gross tumour eradication is the best curative option for any CCA”. Complete surgical resection is the only modality that can cure CCA.

Authors: Thank you very much. The sentence has been changed in:

Complete surgical resection with all gross tumour eradication is the only modality that can cure any CCA.

R: Line 176: “The overall survival after LT at 1 and 3 years was 91% and 81%, respectively [18,27]”. Could the authors clarify from which study were these percentages. Because from the Mayo Clinic’s study, the overall survival was 88% at 1 year and 82 % 5 years after transplantation.

Authors: Apologies for the Typo error. The sentence has been changed and the reference has been edited according to the Reviewer suggestion as follows:

The overall survival after LT at 1 and 3 years was 91% and 81%, respectively [19]. Five-year survival after transplantation and neoadjuvant was 73%, 79% for patients with underlying PSC and 63% for those with de novo CCA [43]

R: Line 188-190: “Nevertheless, several centers performing LT for pCCA adopted the Mayo Clinic protocol as the standard of care whenever complete surgical resection was impossible”. Authors had to modulate the affirmation, and add that this strategy is not recommended in international guidelines.

Authors: Thank you very much. The sentence has been modulated according to the reviewer suggestion as follows.

 However, this strategy is not recommended by International Guidelines

R: A table could be added to compare the different rates of overall survival between different teams studied liver transplantation, compared to surgery.

Authors: Thank you very much. Table 1 has been added following the reviewer suggestion

Authors had to insist on the heterogeneity on chemotherapy protocol studied on neo adjuvant setting before liver transplatation, and the absence of direct comparison.

Authors: Thank you very much. The reviewer suggestion has been incorparated in the following sentence: “On the same note, it is difficult to determine whether LT outcomes in pCCA are partially influenced by the efficacy of neoadjuvant chemoradiotherapy (CRT) as protocols are quite heterogeneous, and there is an absence of direct comparison with surgical resection”.

R: Conclusion:

Lines 264-266: Authors cannot conclude with this affirmation, without any data from a randomized phase 3 trial.

Authors: Thank you very much. According to the reviewer suggestion, the conclusion has been changed: “In several cases, patients diagnosed with pCCA are eligible for surgical resection and sometimes patients can be selected for LT. However, due to the rarity of the disease and lack of phase 3 RCTs, there is often no clear standard pathway to refer to”

R: Lines 288-292: Authors described locoregional treatment; they could develop this paragraph. Authors discussed neo adjuvant treatment before surgery, but in the text, authors did not described studies on surgery. They could add a paragraph with main studies of surgery and the neo adjuvant strategy.

Authors: Thank you very much. The statement about locoregional treatment has been deleted in the present form as there is very weak evidence to support the statement in the general context of the manuscript. Instead, a new paragraph on adjuvant therapy has been added.

Reviewer 2 Report

This manuscript reviewed the latest knowledge about liver trasplantation (LT) for peri-hilar cholangiocarcinoma and covered a clinically relevant topic in this field.   This reviewer has one minor comment. Authors introduced the clinical outcomes of LT from three institutions (Mayo, Toronto, and Michigan). This reviewer considers that a table which summarizes the sample size, treatment periods, efficacy, and safety of LT from three institutions will help readers' understanding.

Author Response

Reviewer #2:

Dear Professor/Doctor,

Thank you very much for your suggestion and recommendation. Please see below the answer to your comment with the realtive editings. All the change have been highlithed in yellow. That has helped to improve the quality of the manuscript.

R: This manuscript reviewed the latest knowledge about liver trasplantation (LT) for peri-hilar cholangiocarcinoma and covered a clinically relevant topic in this field.   This reviewer has one minor comment. Authors introduced the clinical outcomes of LT from three institutions (Mayo, Toronto, and Michigan). This reviewer considers that a table which summarizes the sample size, treatment periods, efficacy, and safety of LT from three institutions will help readers’ understanding.

Authors: Thank you very much. Table 1 has been added following the reviewer suggestion

Reviewer 3 Report

1. The schematic diagram on the right side of Figure 1 should be enlarged.

2. It is suggested to add the content related to adjuvant therapy after liver transplantation.

Author Response

Dear Professor/Doctor,

Thank you very much for your suggestion and recommendation. Please see below the answer to your comment with the realtive editings. All the change have been highlithed in yellow. That has helped to improve the quality of the manuscript.

R: The schematic diagram on the right side of Figure 1 should be enlarged.

Authors: Thank you very much. Fig. 1 has been enlarged as suggested by the reviewer

R: It is suggested to add the content related to adjuvant therapy after liver transplantation.

Authors: Thank you very much. As suggested by the reviewer, a new paragraph on adjuvant therapy has been added.